# T2VTextBench: A Human Evaluation Benchmark for Textual Control in Video Generation Models

## Abstract

Recent advancements in scalable deep architectures and large-scale pretraining have enabled text-to-video generation has achieved unprecedented capabilities in producing high-fidelity, instruction-following content across a wide range of styles, supporting applications in advertising, entertainment, and education. However, these models' ability to render precise on-screen text, such as captions or mathematical formulas, remains largely untested, posing significant challenges for applications requiring exact textual accuracy. In this work, we introduce **T2VTextBench**, the first human-evaluation benchmark dedicated to evaluating on-screen text fidelity and temporal consistency in text-to-video models. Our suite of prompts integrates complex text strings with dynamic scene changes, testing each model's ability to maintain detailed instructions across frames. We evaluate ten state-of-the-art systems, ranging from open-source solutions to commercial offerings, and find that most struggle to generate legible, consistent text. These results highlight a critical gap in current video generators and provide a clear direction for future research aimed at enhancing textual manipulation in video synthesis.

## 1 Introduction

Text-to-video generative AI (Singer et al., 2023; Hong et al., 2023; Wu et al., 2023; Yang et al., 2024) has been a game-changer in real-world applications such as advertising, entertainment, and education purposes. Thanks to recent advances in scalable deep architectures, particularly diffusion models (Ho et al., 2022; Esser et al., 2023) and Transformers (Arnab et al., 2021; Liu et al., 2022), and large-scale training on text-video pairs harvested from the web, such systems currently exhibit unprecedented instruction-following ability and aesthetic quality across a wide variety of styles. State-of-the-art models such as Sora (OpenAI, 2024) and SD Video (Blattmann et al., 2023) are rapidly being adopted as central elements of users' online experiences, gaining widespread attention and impact.

Despite these successes, there is growing concern over text-to-video models' ability to generate accurate on-screen text (Park et al., 2024; Liu et al., 2024a). In many applications, this limitation can have serious consequences. For instance, in advertising, a brand name must be rendered precisely, and in educational videos, explanatory text or mathematical formulas also should appear correctly. While recent research has significantly improved textual manipulation in text-to-image models (Chen et al., 2023; Tuo et al., 2024; Zhu et al., 2024; Peng et al., 2025), the corresponding capability in text-to-video remains largely untested. A systematic evaluation of on-screen text fidelity in video generation is therefore essential both for guiding downstream applications and for uncovering fundamental model limitations.

To address this gap, we propose **T2VTextBench**, a comprehensive benchmark for evaluating textual manipulation in modern text-to-video models. Our benchmark stresses both complex textual content and temporal coherence (consistent rendering across frames), probing each model's ability to follow intricate human instructions. The benchmark also incorporates comprehensive human evaluation, aligning with human preferences and addressing nuances in temporal dynamics. We systematically

evaluate ten leading models, including both open-source and proprietary systems, to cover the latest advances in text-to-video generation.

Our contributions are summarized as follows:

- We propose T2VTextBench, a comprehensive human evaluation on how up-to-date text-to-video generators manipulate on-screen text in videos.

- We analyze the impact of temporal text transformations, geometric, visual, and structural, and demonstrate that current text-to-video models are unstable with these changes.

- We examine the effect of text granularity, finding that models perform well on single words but show significant gaps when generating longer sentences and random individual characters.

- We present a cost analysis evaluating the cost-effectiveness of current models for generating on-screen text.

**Roadmap.** We systematically review the relevant works of this benchmark in Section 2. We describe the details of our proposed T2VTextBench benchmark in Section 3. We show the key assessment results of our benchmark in Section 4. We present some concluding remarks for this paper in Section 5.

## 2 RELATED WORKS

**Visual Text Generation.** Generative visual models have achieved unprecedented success in various real-world applications, producing human-preference-aligned images and videos with high fidelity and aesthetic standards (Ho et al., 2020; Song et al., 2021a;b; Lipman et al., 2023; Hong et al., 2023; Wu et al., 2023; Yang et al., 2024). Despite these advancements, generating accurate and coherent text in visual outputs has increasingly become a significant challenge. One prominent line of research focuses on text rendering in general text-to-image models (Chen et al., 2023; Yang et al., 2023; Zhu et al., 2024; Zhang et al., 2024; Zhao et al., 2024). For instance, TextDiffuser (Chen et al., 2023) determines text layout using a Transformer model and generates textual images with diffusion models conditioned on both textual prompts and text layouts, while GlyphControl (Yang et al., 2023) enhances text rendering capabilities. Next, SceneVTG (Zhu et al., 2024) leverages the strong reasoning ability of multimodal LLMs to suggest desirable text layouts and content at various scales and levels for diffusion model conditioning.

Another important line of research focuses on typography design and generation using generative AI. These models generate specific textual icons or word art using specially designed architectures, with many early works emphasizing the generation of static images (He et al., 2023; Xiao et al., 2024; Park et al., 2024; Choi et al., 2024). Recently, KineTy (Park et al., 2024) introduced a method for generating textual videos with diverse visual effects, employing zero-convolution guidance to control text visibility and glyph loss to ensure readability. Despite previous successes in text manipulation within general text-to-image models and typography-focused models, these approaches either overlook the temporal dynamics of multi-frame videos or fail to generate general-purpose videos beyond typography. Consequently, there remains a substantial gap in text manipulation within text-to-video models, making it highly desirable to benchmark current progress and highlight future research directions.

Several pioneering studies have explored text generation in text-to-video models. For example, Wan2.1 (Alibaba, 2025) enhances text-to-video models' intrinsic text generation capabilities through data curation and large-scale training, incorporating hundreds of millions of synthetic text images along with OCR-annotated real image-text pairs. Meanwhile, Text-Animator (Liu et al., 2024a) proposes a plug-and-play approach to improve the 3D-Unet in existing text-to-video generation models, injecting text embeddings via ControlNet and incorporating a camera control module to embed perspective-related pose information. However, while these models excel at generating relatively short text in videos, they may not adequately address complex temporal dynamics involving motion or text organization. In contrast, our benchmark considers these factors and provides an in-depth evaluation.

**Benchmarking Text-to-Video Generative Models.** Due to the widespread real-world impact of text-to-video models, evaluating their capabilities has become a fundamental research topic. Prior work on benchmarking these generative models has addressed many important aspects, including but not limited to the composition of different properties (Feng et al., 2024; Sun et al., 2024), video fidelity (Liu et al., 2023), temporal coherence (Liao et al., 2024; Ji et al., 2024), object counting (Guo et al., 2025a; Cao et al., 2025b), physical constraint adherence (Meng et al., 2024; Guo et al., 2025b), and storytelling (Bugliarello et al., 2023). Specifically, (Liu et al., 2023) proposed a fine-grained evaluation benchmark for text-to-video models, considering three fundamental factors: major content, controllable attributes, and prompt complexity, with manual evaluation conducted on four mainstream text-to-video models. EvalCrafter (Liu et al., 2024b) introduces a comprehensive benchmark featuring an exhaustive set of prompts, 17 automated objectives, and coverage both image-to-video and text-to-video models. VBench (Huang et al., 2024a;b) evaluates video quality and prompt consistency across 16 human-aligned dimensions, each with tailored prompts and metrics validated against human preferences. Despite the effectiveness of previous benchmarks, most focus on concrete entities (e.g., humans, real objects) while overlooking the ability to generate text and maintain temporal text consistency, which motivates the exploration in our paper. The proposed benchmark inspires a wide range of future works, such as limitation of deep visual architectures (Chen et al., 2025b; Li et al., 2025; Hu et al., 2024a; 2025a;b; Chen et al., 2025c; 2024; Li et al., 2024; Ke et al., 2025), theory in diffusion models (Hu et al., 2024b; 2025c; Chen et al., 2025a; Cao et al., 2025a), novel diffusion model architectures (Wen et al., 2024; Wang et al., 2024b; 2023; 2024a; Liang et al., 2025).

## 3 THE T2VTEXTBENCH BENCHMARK

We present our proposed T2VTextBench benchmark in this section. We show the baseline models in Section 3.1. We describe the detailed setting of benchmark prompts in Section 3.2. We illustrate our evaluation standard in Section 3.3.

### 3.1 BASELINE MODELS

Table 1: **Overview of 10 Evaluated Text-to-Video Models in Our Benchmark.**

| Model Name | Year | Organization | # Params | Open |
|:---:|:---:|:---:|:---:|:---:|
| SD Video (Blattmann et al., 2023) | 2023 | Stability AI | 1.4B | Yes |
| Kling (Kling, 2024) | 2024 | Kuai | N/A | No |
| Dreamina (ByteDance, 2024) | 2024 | ByteDance | N/A | No |
| Qingying (Zhipu, 2024) | 2024 | Zhipu | 5B | Yes |
| Sora (OpenAI, 2024) | 2024 | OpenAI | N/A | No |
| Mochi-1 (Genmo, 2024) | 2024 | Genmo | 10B | Yes |
| LTX Video (HaCohen et al., 2024) | 2024 | Lightricks | 2B | Yes |
| Hailuo (MiniMax, 2025) | 2025 | MiniMax | N/A | No |
| Wan 2.1 (Alibaba, 2025) | 2025 | Alibaba | 14B | Yes |
| Pika 2.2 (Pika, 2024) | 2025 | Pika Labs | N/A | No |

In this paper, we evaluate a broad selection of modern text-to-video generators that have been publicly available from 2023 to 2025. This model selection reflects the latest advancements in text-to-video systems and reliably highlights their inherent limitations in handling complex textual content within videos. Specifically, we assess 10 models, including both commercial and open-source generators. Basic model information is provided in Table 1.

To generate videos, this benchmark adopts the lowest accessible resolution of these generative models, typically 720p, to reach a balance between video quality and textual accuracy. We maintain a 16:9 width-height ratio and limit the length of videos to a short span, usually around 4 seconds, to focus the assessment on core textual behaviours. Additional details regarding implementation can be found in Appendix A.

## 3.2 BENCHMARK PROMPTS

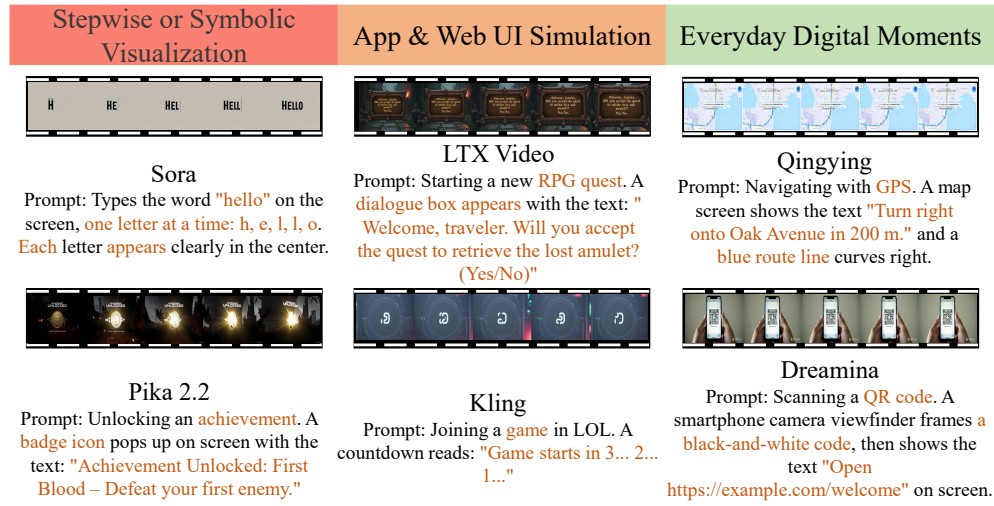

Figure 1: **T2V Model Generate Text Result Across Different Measures**.

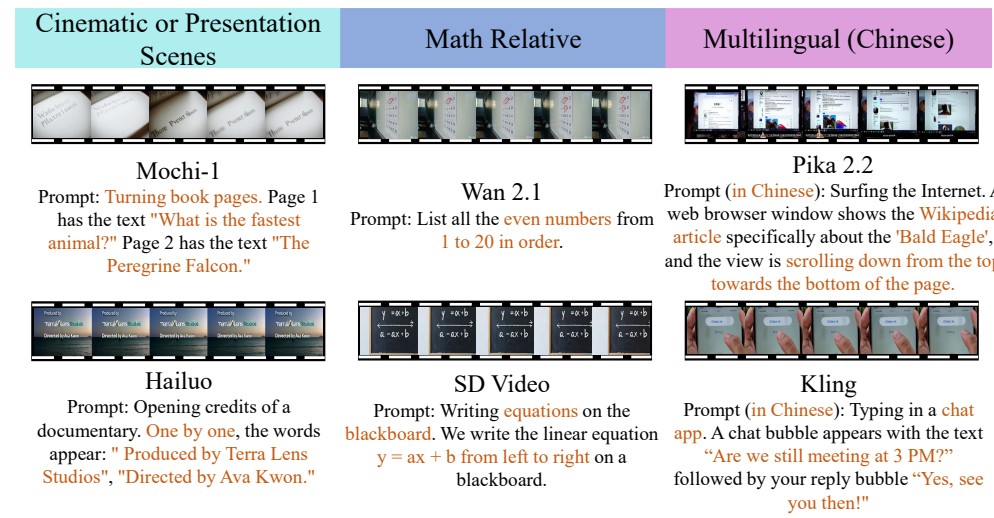

Figure 2: **T2V Model Generate Text Result Across Different Measures**.

In this project, we design a prompt suite to assess the text generation capabilities of text-to-video models under complex temporal dynamics. Our evaluation scenarios are grounded in real-world settings, addressing both text manipulation and contextual coherence. Specifically, we consider six types of prompts: Stepwise or Symbolic Visualization, App & Web UI Simulation, Everyday Digital Moments, Cinematic or Presentation Scenes, Math-Related, and Multilingual (Chinese). Each category consists of 8 prompts, and with additional ablation study prompts included in the overall results, we have a complete suite of 73 prompts for each generative model. Example prompts and corresponding videos can be found in Figure 1 and Figure 2.

## 3.3 EVALUATION STANDARD

Inspired by the prior success of FETV (Liu et al., 2023) and VideoPhy (Bansal et al., 2025), we employ a fully human-evaluation approach in this paper to align with human preferences and account for the inherent nuances of temporal dynamics and text within specific contexts. We invite three undergraduate or graduate students with a general level of expertise in AI, who independently

examine each output video and assign it to one of the predefined accuracy levels. Specifically, we introduce a 0-1 point scale evaluation standard for assessing the ability of text-to-video models to generate text:

- **0 (Poor)**: The model does not understand the prompt requirements at all, producing generated text that is completely gibberish or irrelevant.
- **0.25 (Fair)**: The model partially understands the prompt's textual requirements, but only a small portion of the generated text (less than 50%) is correct or recognizable, while the majority of the content remains significantly incorrect or missing.
- **0.5 (Good)**: The model demonstrates a solid understanding of the prompt's textual requirements, with the majority (50% - 80%) of the generated text being correct and capturing the core content, though some errors or omissions persist.
- **1 (Excellent)**: The model accurately comprehends and executes the textual requirements of the prompt, generating text that is either almost entirely correct or contains only a few minor, insignificant flaws.

This evaluation method grants full credit for entirely correct generations while also awarding partial points for near-accurate cases. For each model, we compute an overall text-generation score by averaging the scores across all prompts and annotators. This final score is then used to rank the models accordingly.

## 4 EXPERIMENTS

In Section 4.1, we show the overall assessment results on generating textual contents. In Section 4.2, we show some observations of text-to-video models' performance changes under text spatial or colour transformations. In Section 4.3, we present the influence of prompts with various levels of text organization on generation quality. In Section 4.4, we show the pricing analysis.

### 4.1 OVERALL EVALUATION RESULTS

Table 2: **Overall Results Within Various Prompt Categories.**

| Model | Stepwise | App/Web UI | Digital Moments | Cinematic | Math | Multilingual | Avg. Score |
|---|---|---|---|---|---|---|---|
| Kling | 0.03 | 0.00 | 0.00 | 0.00 | 0.00 | 0.00 | 0.01 |
| Dreamina | 0.06 | 0.19 | 0.06 | 0.19 | 0.09 | 0.09 | 0.11 |
| Qingying | 0.13 | 0.13 | 0.25 | 0.22 | 0.06 | 0.25 | 0.17 |
| Mochi-1 | 0.13 | 0.13 | 0.25 | 0.25 | 0.16 | 0.28 | 0.19 |
| LTX Video | 0.19 | 0.19 | 0.25 | 0.22 | 0.22 | 0.16 | 0.20 |
| Wan 2.1 | 0.28 | 0.44 | 0.38 | 0.34 | 0.31 | 0.25 | 0.33 |
| SD Video | 0.28 | 0.34 | 0.34 | 0.38 | 0.31 | N/A | 0.33 |
| Hailuo | 0.25 | 0.47 | 0.34 | 0.50 | 0.25 | 0.31 | 0.35 |
| Pika 2.1 | 0.41 | 0.28 | 0.44 | 0.41 | 0.31 | 0.34 | 0.36 |
| Sora | 0.44 | 0.50 | 0.47 | 0.28 | 0.28 | 0.25 | 0.37 |

In this experiment, we compared all the models previously listed in Table 1 based on their text manipulation capabilities across six prompt categories. Specifically, the table presents all the categories from Section 3.2 with the following acronyms: Stepwise or Symbolic Visualization (Stepwise), App & Web UI Simulation (App & Web UI), Everyday Digital Moments (Digital Moments), Cinematic or Presentation Scenes (Cinematic), Math-Related (Math), and Multilingual (Chinese) (Multilingual). The results are presented in Table 2.

According to the table, all models exhibit noticeable failures in generating videos with textual content. The highest average score is reported for Sora, which is only 0.37. Another striking result is from Kling, which has almost no text generation ability and fails entirely in all categories except for Stepwise visualization, where it achieves only a 0.01 average score. This highlights the limitations of current text-to-video models in generating coherent and accurate textual content.

**Observation 4.1.** *Recent advances in text-to-video generative models do not adequately support the generation of textual content, as all models have an average score below 0.4, indicating a consistent failure.*

Another observation concerns the large variance between both models and prompt categories. First, a model's performance can differ significantly across prompt categories. For instance, Sora reports a score of 0.5 for App/Web UI prompts, but as low as 0.28 for Cinematic prompts. Meanwhile, Hailuo achieves 0.5 in Cinematic scenarios, but its score drops to 0.25 for mathematical text prompts. Across different models, Sora and Pika 2.1 have overall accuracy above 0.35, while Dreamina and Kling score below 0.12, demonstrating a large variance between models. Therefore, we observe the following:

**Observation 4.2.** *The variance between different models and prompt categories is significant, highlighting the instability of current text-to-video models in textual generation tasks.*

**Qualitative Study.** To further verify our observations, we carefully examine some video examples in Figure 1 and Figure 2 qualitatively. In Figure 1, Kling produces only random symbols with no recognizable letters, Dreamina cannot understand the prompt accurately and generates the prompt partially, and Qingying generates overlapping glyphs, making it less readable. Likewise, in Figure 2, Mochi-1 generates meaningless strings unrelated to the prompt, and Wan 2.1 outputs distorted numerals that do not match the target text. These failures in both figures, confirm the uniformly poor text-rendering ability of current models in Observation 4.1.

Besides, in Figure 1, Sora successfully follows a multi-step English prompt with clear, coherent letters, while Kling fails completely and LTX Video produces partially blurred characters. In Figure 2, Hailuo correctly renders the Chinese prompt with only minor glyph errors, whereas SD Video replaces the target formula with an incorrect equation. These differences across the same and different prompts highlight the instability and uneven performance of text-to-video models as shown in Observation 4.2.

## 4.2 IMPACT OF TEXT TRANSFORMATIONS

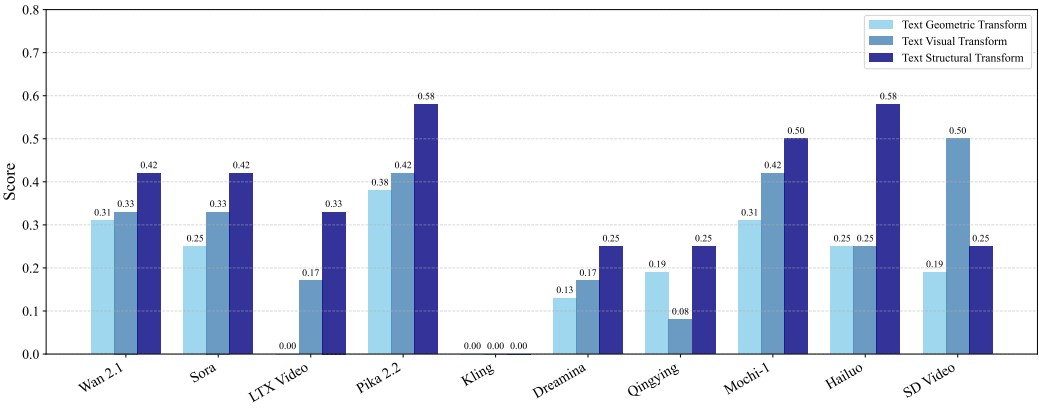

Figure 3: **Ablation Study on Text Generation under Different Transformations**.

In this study, we explore the capability of text-to-video models to generate complex geometric, visual, and structural transformations of text, thereby evaluating their ability to follow more intricate human instructions. Specifically, we use three types of prompts (see Figure 4 for example inputs and outputs):

- **Geometric**: Prompts that require simple geometric transformations of the text, such as translation or rotation.
- **Visual**: Prompts that specify changes in visual properties of the text, such as color shifts, fading, or blinking.
- **Structural**: Prompts that request structural transformations of the text, for example, rendering the text as a waving flag or a rainbow.

The results of this ablation study are shown in Figure 3. First, we observe that all models exhibit substantial room for improvement. For example, the best-performing model, Pika 2.2, achieves

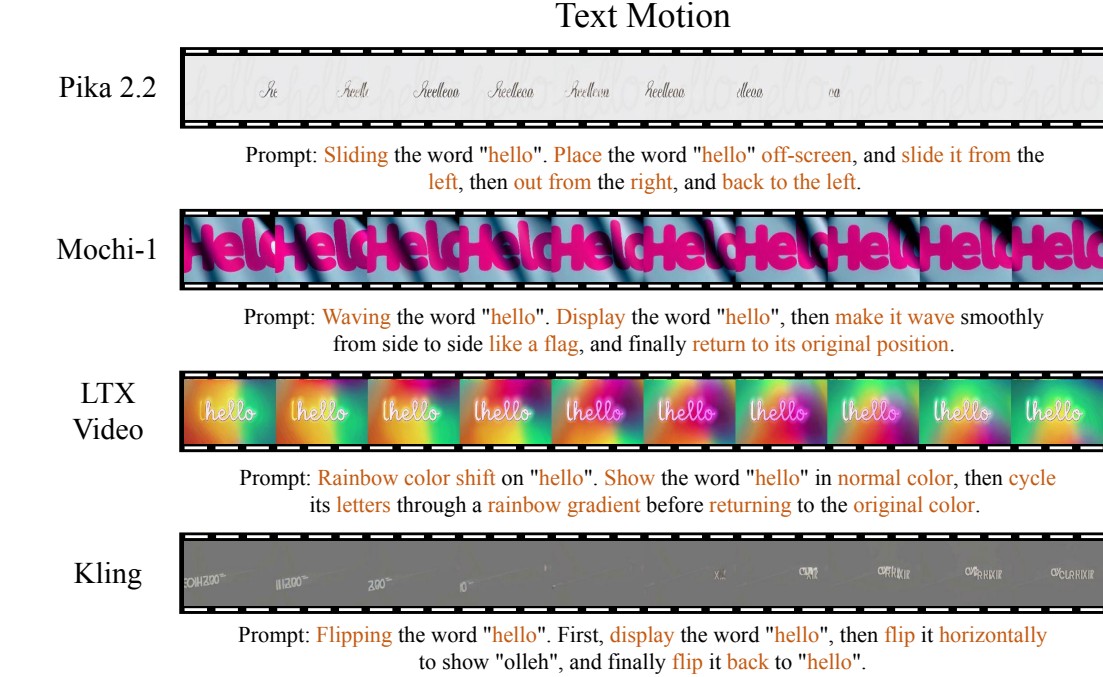

Figure 4: **Case Study on Text Transformations**.

scores of 0.38, 0.42, and 0.58 on geometric, visual, and structural transforms, respectively. As in earlier experiments, Kling fails on all three types of prompts, scoring zero across the board. Across all models and classes, no score exceeds 0.58, and most scores lie between 0.2 and 0.45. Thus, we have the following observation:

**Observation 4.3.** *There remains a significant performance gap on all three text-motion classes, indicating a clear limitation of current text-to-video models in generating temporal variations of textual content.*

We also observe that prompt difficulty varies by category. Figure 3 reveals a consistent difficulty ranking, except for Qingying and SD Video, where geometric transforms yield the lowest scores, visual transforms perform moderately better, and structural transforms receive the highest scores. For example, LTX Video fails entirely on geometric prompts, scores 0.17 on visual transforms, and peaks at 0.33 on structural transforms. While the category gaps are modest overall, some models like LTX Video show dramatic variation across classes. Conversely, models such as Wan 2.1 exhibit consistent performance, with only a 0.10 difference between its lowest (0.31 on geometric) and highest (0.42 on structural) scores. Thus, we have the following observation:

**Observation 4.4.** *With few exceptions, most models consistently score lowest on geometric transformations, achieve moderate performance on visual changes, and perform best on structural transformations. The size of this category-wise performance gap varies considerably between models.*

**Qualitative Study.** To further demonstrate the widespread failure of text-to-video models in generating transformed textual content, we highlight several illustrative bad cases in Figure 4. All four models fail to render the word "hello" correctly, producing outputs such as "lhello" in LTX Video, "helo" in Mochi-1, or entirely garbled text in Kling. In addition to incorrect textual rendering, the transformation instructions are also ignored. For example, LTX Video changes only the background color and does not alter the text color as specified.

### 4.3 IMPACT OF TEXT RANDOMNESS

In this study, we consider an in-depth setting that primarily aims to examine whether text-to-video diffusion models truly understand text manipulation or merely memorize text snippets from their

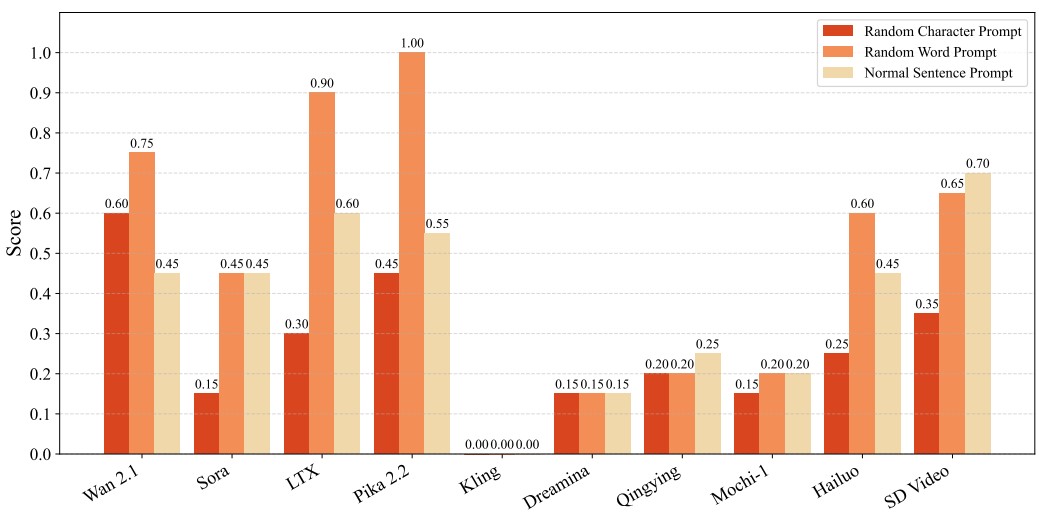

Figure 5: **Ablation Study of Prompts with Different Levels of Text Randomness**.

training data. Specifically, we explore different levels of randomness in our textual prompts and introduce a three-level prompt suite (see Figure 6 for prompt and video examples):

- **Normal Sentence**: All sentences generated at this level are meaningful and recognizable.
- **Random Word**: These prompts contain sentences composed of random and uncommon words with no actual meaning.
- **Random Character**: All characters in this level are completely random and meaningless.

Our results for textual randomness can be found in Figure 5. First, we observe that in all three categories, despite an outlier observed in Wan 2.1, random character prompts yield the worst performance. For instance, in LTX Video, the result is 0.9 for random word prompts, while the counterpart for random character prompts is only 0.3. The second-best performing category is the random word prompt, as normal sentence prompts perform significantly worse than random word prompts in Wan 2.1 (0.75 to 0.45), LTX (0.9 to 0.6), Pika 2.2 (1 to 0.55), and Hailuo (0.6 to 0.45). In other models, the gap between random word and normal sentence prompts is either negligible or nonexistent. Thus, we observe the following:

**Observation 4.5.** *Across all models, random word prompts yield the best performance, normal sentence prompts show moderate performance, and random character prompts perform the worst. This suggests that text-to-video models primarily memorize textual training data at the word level, while their ability to control sentence structure and individual characters still requires improvement.*

**Qualitative Study.** We further present a case study in Figure 6 to support our quantitative findings. Specifically, we find that Pika 2.2 almost perfectly generated the correct text and performed excellently in terms of text continuity, which aligns with its strong performance in Figure 5. Although Dreamina correctly generated the scene, it included text that was irrelevant to the task. Kling completely failed to generate the correct text and was unable to follow the task instructions. These results match their low quantitative performance.

## 4.4 PRICING ANALYSIS

Pricing analysis plays a crucial role in identifying the most cost-efficient models, enabling users to make informed, cost-effective decisions while ensuring the desired output quality. We compare the costs and performance scores of text-to-video models based on the 73 prompts discussed in Section 3.2. By evaluating each model's average cost and average performance score, we assess the trade-offs between expense and output quality. Specifically, the SD Video model supports only 65 prompts due to its inability to handle Chinese input.

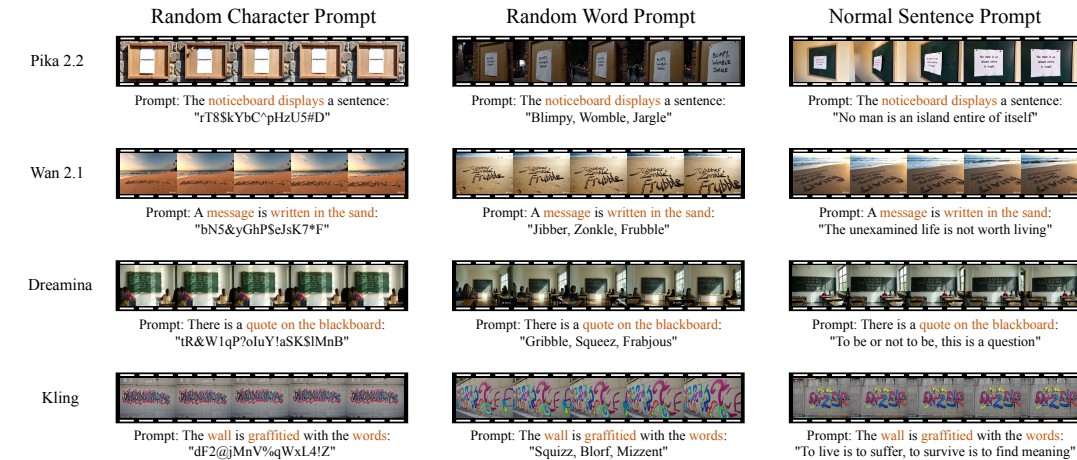

Figure 6: **Case Study on Text Randomness**.

Table 3: **Cost and Score Across Different Models.**

| Model | Prompt Number | Total Cost($) | Avg. Cost($) | Avg. Score |
|---|---|---|---|---|
| Kling | 73 | 15.84 | 0.22 | 0.01 |
| Dreamina | 73 | 3.79 | 0.05 | 0.13 |
| Qingying | 73 | 5.00* | 5.00* | 0.18 |
| Mochi-1 | 73 | 9.12 | 0.13 | 0.22 |
| LTX Video | 73 | 4.39 | 0.06 | 0.28 |
| Sora | 73 | 20.00* | 20.00* | 0.36 |
| Hailuo | 73 | 21.90 | 0.30 | 0.37 |
| SD Video | 65 | 17.30 | 0.24 | 0.38 |
| Wan 2.1 | 73 | Free | Free | 0.39 |
| Pika 2.2 | 73 | 20.00 | 0.27 | 0.44 |

The results of the pricing analysis are shown in Table 3. First, we observe a positive correlation between a model's pricing and text generation quality. For example, Dreamina and Mochi-1 are among the most affordable models, with average costs of 0.05 and 0.13, respectively, but their average performance scores are relatively low, at 0.13 and 0.22. In contrast, higher-priced models like Hailuo and Pika 2.2, priced at 0.30 and 0.27, achieve much better results, with scores of 0.37 and 0.44. Thus, we observe the following:

**Observation 4.6.** *Apart from a few outliers, higher-cost models tend to deliver better generation quality, reflecting a general positive correlation between price and performance.*

## 5 CONCLUSION

In this work, we introduced T2VTextBench to systematically evaluate on-screen text fidelity in modern text-to-video generation models, addressing a critical gap in their ability to render precise textual content. Our extensive human-evaluation of ten leading systems revealed a consistent failure to generate accurate on-screen text (all models scoring below 0.43), significant instability across prompt categories and model architectures, and clear limitations under temporal text transformations, particularly geometric ones. We further showed that while models handle single words reasonably well, their performance degrades sharply on longer sentences and arbitrary character sequences, indicating reliance on memorized word-level patterns. Finally, our cost analysis uncovered a positive correlation between API cost and generation quality, with Wan 2.1 emerging as the most cost-effective solution. These findings highlight the need for future research on integrating explicit text modeling, enhancing temporal coherence, and developing more efficient architectures to support reliable textual manipulation in video.

## ETHIC STATEMENT

This paper does not involve human subjects, personally identifiable data, or sensitive applications. We do not foresee direct ethical risks. We follow the ICLR Code of Ethics and affirm that all aspects of this research comply with the principles of fairness, transparency, and integrity.

## REPRODUCIBILITY STATEMENT

We ensure the reproducibility of our empirical findings. For all experiments, we describe the sources of the LLM models, datasets, evaluation metrics, and experiment setup in the main text. Several example prompts used are also provided to support the reproducibility of our results.

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

# Appendix

**Roadmap.**   In Section A, we describe the detailed implementation settings for our selected baseline models.  In Section B, we outline an impact statement on the societal implications of this benchmark. In Section C, we show additional video samples under different prompts.

## A  BASELINE DETAILS

In this subsection, we extend the basic model information in Table 1 with extra details of these text-to-video generation models. Specifically, our additional experimental settings are shown as follows:

- **Kling** (Kling, 2024): For Kling's four different versions, we select a recent version, Kling 1.6, within its standard generation mode. We use its basic standard generation mode with no creative parameters.  It does not offer camera movement options.  We use the default random seed setting in the generation processes.

- **Wan 2.1** (Alibaba, 2025): We adopt the fast generation mode for Wan 2.1 with an aspect ratio of 16:9. We use the default setting for prompt input and Inspiration Mode, and do not add sound effects in the generated videos.

- **Sora** (OpenAI, 2024): Sora is a proprietary text-to-video generator introduced by OpenAI in 2024.  It operates in a single mode and supports output resolutions of 480p, 720p and 1080p, with aspect ratios of 16:9, 1:1, and 9:16.  We generate 5-second videos with a 30 FPS refresh rate. This model can generate four videos simultaneously for a single prompt. After reaching its daily generation limit, it will slow down its generative process.

- **Mochi-1** (Genmo, 2024): Mochi-1 generates 5-second videos at 24 FPS. We use the default setting for prompt hints and seed functions.  This model renders each batch of videos for three minutes, and each batch contains two video samples. After several generations, this model will slow down its generation process.

- **LTX Video** (HaCohen et al., 2024):  Different from other models, this model does not support our standard 720p setting, and only supports a 768×512 (512p) resolution, with a 24 FPS refresh rate. For extra settings, such as scene settings or style settings, we use the default version.

- **Pika 2.2** (Pika, 2024):  In this model, we use the basic setting of Pika 2.2 without additional features such as PikaFrames and PikaEffects.  We default the negative prompt and seed configurations. This model generates four videos simultaneously, each requiring approximately 30 seconds, and enables prompt copying and editing with a single click.

- **Dreamina** (ByteDance, 2024): For Dreamina, we adopt its Video S2.0 version without its prompt enhancements from external LLMs. All the generated videos are by default in 24 FPS, and there are no other refresh rate settings.

- **Qingying** (Zhipu, 2024): Qingying serves as the commercial version of CogVideo (Hong et al., 2023) and CogVideoX (Yang et al., 2024). We use its fast generation model, with 5-second and 30 FPS sound-free video settings.  For all the advanced options on style, emotion and camera movements, we adopt its default setting.

- **Hailuo** (MiniMax, 2025): Different from other models, Hailuo's only duration setting is 6 seconds and 24 FPS, and we adopt this video length.  For text conversion, we use the T2V-01-Director base model.

- **Stable Video Diffusion** (Blattmann et al., 2023): We use the default generation setting on video length for Stable Video Diffusion, which is 4 seconds.

## B  IMPACT STATEMENT

The paper discusses potential positive impacts on generating accurate and temporally coherent text in text-to-video models, which may bring social good across entertainment, education, and other domains. Its comprehensive evaluation framework will accelerate the development of more reliable and user-friendly video generation systems. We do not foresee significant negative societal impacts.

Although enhanced text manipulation could facilitate the creation of realistic but misleading video content, existing safety barriers and responsible deployment practices for large models help mitigate this risk.

## C  VIDEO EXAMPLES

In this subsection, we show additional video samples generated with this benchmark's extensive provided prompts. The results are outlined in Figure 7 to Figure 16. Each result highlights outputs from a single generative model across six distinct scenarios. To illustrate temporal changes, ten key frames were extracted from each clip. These chosen examples correspond to the full set of experimental cases outlined in Section 4.

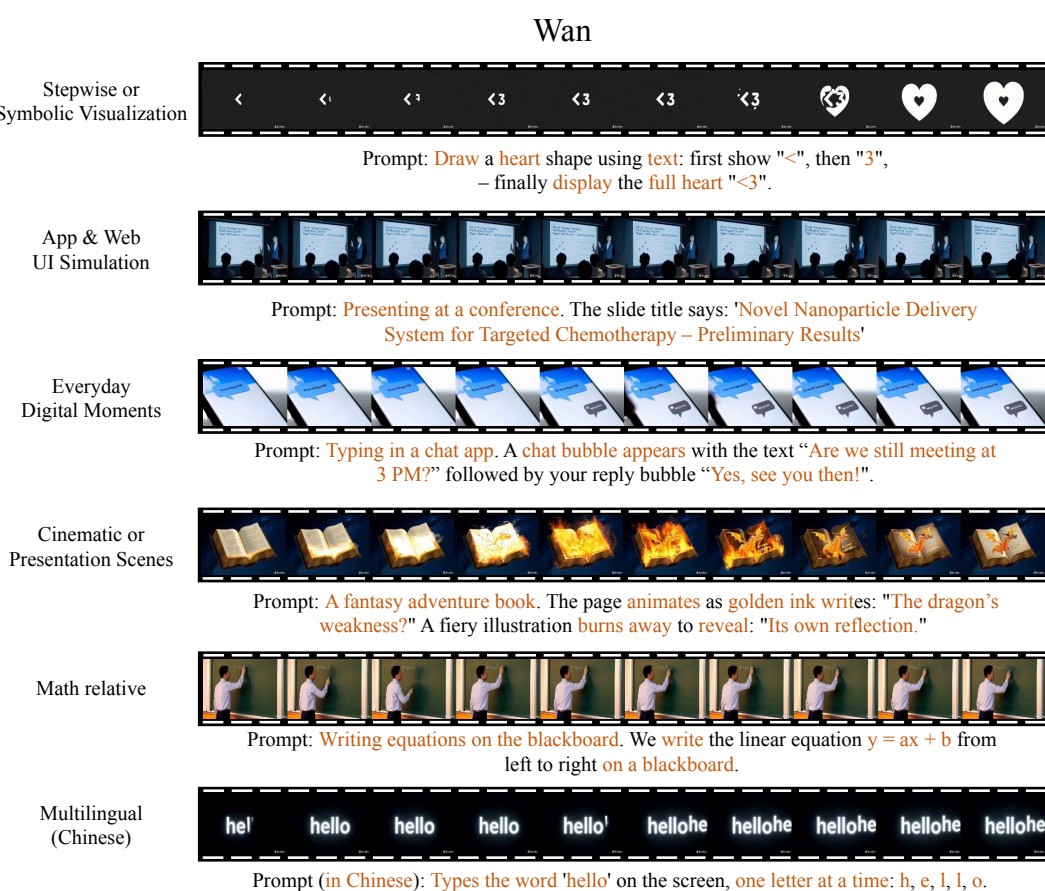

Figure 7: **Results of Videos Generated by Wan**.

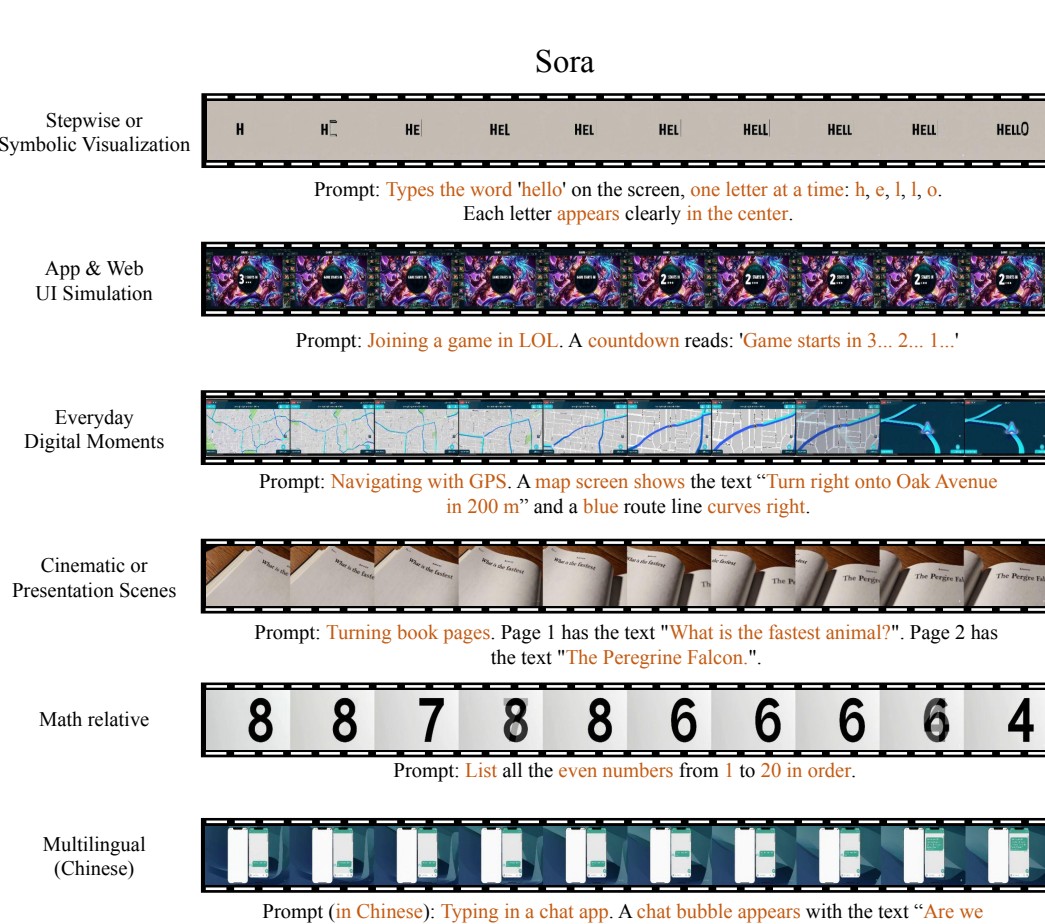

Figure 8: **Results of Videos Generated by Sora.**

LTX Video

Stepwise or
Symbolic Visualization

Prompt: Show the process of typing "WOW" one letter at a time: W, O, W.

App & Web
UI Simulation

Prompt: Using a calculator app. You type "45 × 12" and the display
shows "540" immediately.

Everyday
Digital Moments

Prompt: Typing in a chat app. A chat bubble appears with the text "Are we still meeting at
3 PM?" followed by your reply bubble "Yes, see you then!".

Cinematic or
Presentation Scenes

Prompt: Opening credits of a documentary. One by one, the words appear:
"Produced by TerraLens Studios", "Directed by Ava Kwon".

Math relative

Prompt: Write out the multiplication table for numbers 1 through 5,
arranging the results in a grid.

Multilingual
(Chinese)

Prompt (in Chinese): Joining a game in LOL. A countdown reads:
'Game starts in 3... 2... 1...'

Figure 9: **Results of Videos Generated by LTX Video.**

## Pika 2.2

Stepwise or
Symbolic Visualization

Prompt: Show the ticking process of a digital clock: first display "12:00",
then "12:01", followed by "12:02".

App & Web
UI Simulation

Prompt: Opening scene of a thriller film. On-screen text appears: "Chicago, 1987".

Everyday
Digital Moments

Prompt: Compare two product prices: 'Notebook – $5.99 at Store A' vs.
'Notebook – $4.50 at Store B'. Highlight the cheaper one.

Cinematic or
Presentation Scenes

Prompt: A superhero origin story. Comic book text "POW!" and "BAM!"
flash as the screen cracks: "Power activated."

Math relative

Prompt: Draw a triangle with angles 60°, 60°, and 60°, and label each angle.

Multilingual
(Chinese)

Prompt (in Chinese): Surfing the Internet. A web browser window shows the Wikipedia article specifically
about the 'Bald Eagle',and the view is scrolling down from the top towards the bottom of the page.

Figure 10: **Results of Videos Generated by Pika 2.2**.

## Kling

| | |
|---|---|
| Stepwise or Symbolic Visualization | |

Prompt: Show the process of typing "WOW" one letter at a time: W, O, W.

App & Web UI Simulation

Prompt: Signing a PDF document. You click the "Signature" button, draw your signature "J. Smith" in the box, and save the document.

Everyday Digital Moments

Prompt: Activating an alarm on a phone. The screen shows: 'Alarm set for 7:00 AM'. The alarm rings, and the screen updates to: 'Wake up! 7:00 AM'

Cinematic or Presentation Scenes

Prompt: Neon city news. Glowing blue text scrolls across skyscraper screens: "Today's precipitation will contain trace nanobots - umbrellas recommended".

Math relative

Prompt: Use square units to show the area of a rectangle with length 4 and width 3. Draw and count the squares.

Multilingual (Chinese)

Prompt (in Chinese): Typing in a chat app. A chat bubble appears with the text "Are we still meeting at 3 PM?" followed by your reply bubble "Yes, see you then!".

Figure 11: **Results of Videos Generated by Kling**.

Dreamina

Stepwise or Symbolic Visualization

Prompt: Unlocking an achievement. A badge icon pops up on screen with the text: 'Achievement Unlocked: First Blood – Defeat your first enemy.'

App & Web UI Simulation

Prompt: Reading a box plot label. The chart title says: 'Gene Expression Levels – Control vs Treated – Median Shift: +2.4 units'

Everyday Digital Moments

Prompt: Navigating with GPS. A map screen shows the text "Turn right onto Oak Avenue in 200 m" and a blue route line curves right.

Cinematic or Presentation Scenes

Prompt: A sports highlight reel. Bold text slides in: "Game 7 – 3 seconds left..." The scoreboard updates: "108-107" as the crowd erupts.

Math relative

Prompt: Show a math word problem: 'Sara has 3 apples. She buys 2 more. How many does she have now?' Then display: 'Answer: 5 apples'.

Multilingual (Chinese)

Prompt (in Chinese): Use square units to show the area of a rectangle with length 4 and width 3. Draw and count the squares.

Figure 12: **Results of Videos Generated by Dreamina**.

## Qingying

**Stepwise or Symbolic Visualization**

Prompt: Display the days of the week, one by one: Monday, Tuesday, Wednesday...

**App & Web UI Simulation**

Prompt: Starting a new RPG quest. A dialogue box appears with the text: 'Welcome, traveler. Will you accept the quest to retrieve the lost amulet? (Yes/No)'

**Everyday Digital Moments**

Prompt: Posting a tweet. The Twitter compose box contains "Just tried the new café on Main Street—highly recommend!" and the "Tweet" button is clicked.

**Cinematic or Presentation Scenes**

Prompt: Turning book pages. Page 1 has the text "What is the fastest animal?".
Page 2 has the text "The Peregrine Falcon.".

**Math relative**

Prompt: Write out the multiplication table for numbers 1 through 5,
arranging the results in a grid.

**Multilingual (Chinese)**

Prompt (in Chinese): Converting units: "Convert 100 cm to meters." Display the
calculation: "100 cm = 100 ÷ 100 = 1 meter".

Figure 13: **Results of Videos Generated by Qingying**.

## Mochi-1

Stepwise or Symbolic Visualization

Prompt: Show a progress update: "Task 1: Starting...", then 20%, 50%, 80%, and finally "Task Completed!" when the progress hits 100%.

App & Web UI Simulation

Prompt: Signing a PDF document. You click the "Signature" button, draw your signature "J. Smith" in the box, and save the document.

Everyday Digital Moments

Prompt: Scanning a QR code. A smartphone camera viewfinder frames a black-and-white code, then shows the text "Open https://example.com/welcome" on screen.

Cinematic or Presentation Scenes

Prompt: A sports highlight reel. Bold text slides in: "Game 7 – 3 seconds left..." The scoreboard updates: "108-107" as the crowd erupts.

Math relative

Prompt: Use square units to show the area of a rectangle with length 4 and width 3. Draw and count the squares.

Multilingual (Chinese)

Prompt (in Chinese): Presenting at a conference. The slide title says: 'Novel Nanoparticle Delivery System for Targeted Chemotherapy – Preliminary Results'

Figure 14: **Results of Videos Generated by Mochi-1.**

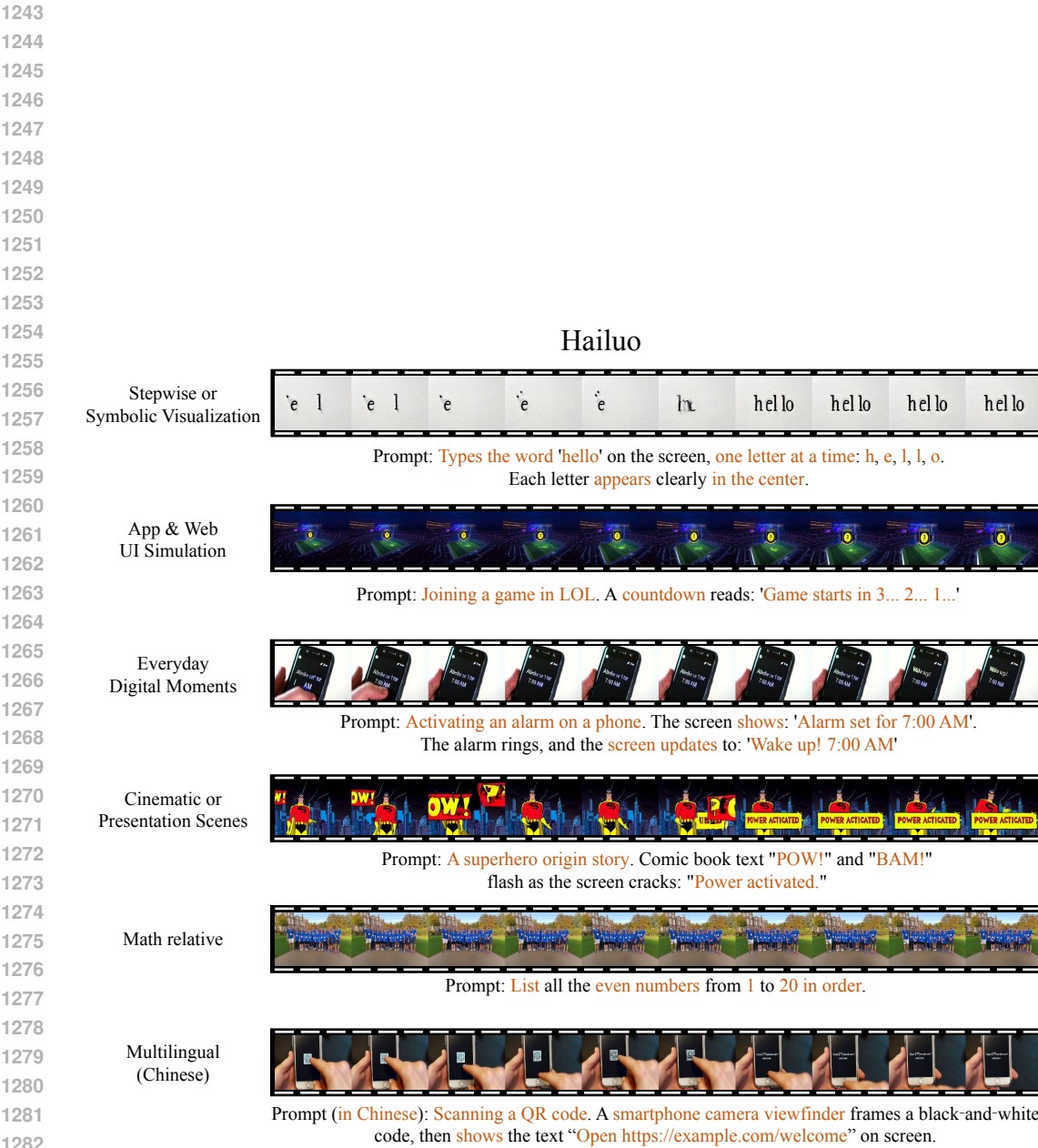

Figure 15: **Results of Videos Generated by Hailuo**.

SD Video

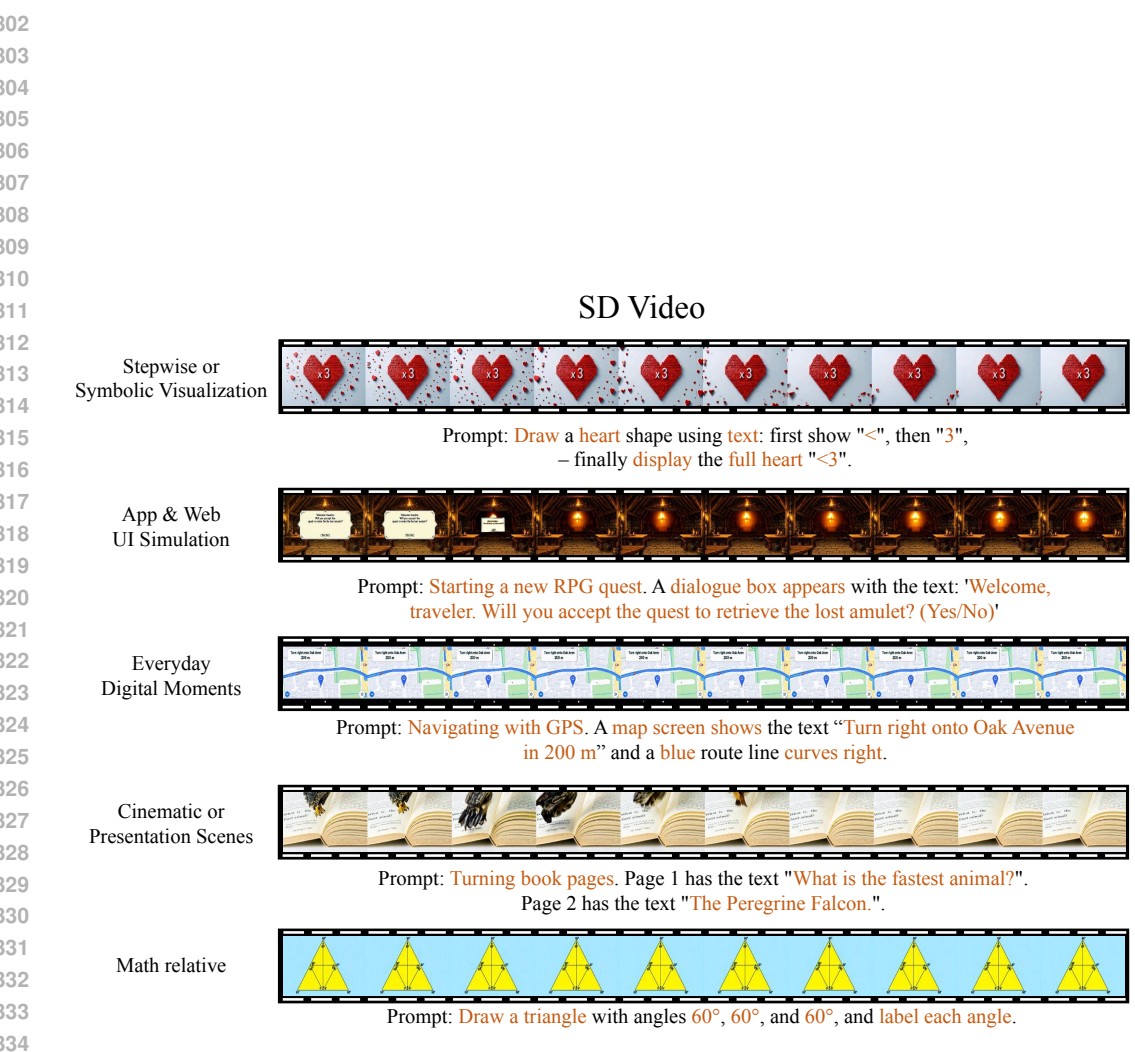

Stepwise or Symbolic Visualization

Prompt: Draw a heart shape using text: first show "<", then "3",
– finally display the full heart "<3".

App & Web UI Simulation

Prompt: Starting a new RPG quest. A dialogue box appears with the text: 'Welcome,
traveler. Will you accept the quest to retrieve the lost amulet? (Yes/No)'

Everyday Digital Moments

Prompt: Navigating with GPS. A map screen shows the text "Turn right onto Oak Avenue
in 200 m" and a blue route line curves right.

Cinematic or Presentation Scenes

Prompt: Turning book pages. Page 1 has the text "What is the fastest animal?".
Page 2 has the text "The Peregrine Falcon.".

Math relative

Prompt: Draw a triangle with angles 60°, 60°, and 60°, and label each angle.

Figure 16: **Results of Videos Generated by SD Video.**

## LLM USAGE DISCLOSURE

LLMs were used only to polish language, such as grammar and wording. These models did not contribute to idea creation or writing, and the authors take full responsibility for this paper's content.

