# OpenReview forum: "T2VTextBench: A Human Evaluation Benchmark for Textual Control in Video Generation Models"
_ICLR.cc/2026/Conference — ICLR 2026 Conference Withdrawn Submission_

### Official Review · Reviewer_tb65 · 2025-10-27

**Soundness:** 2
**Presentation:** 3
**Contribution:** 2
**Rating:** 2
**Confidence:** 4

**Summary:**

The paper introduces T2VTextBench, a benchmark designed to evaluate the quality of text generated within videos. Human raters assign discrete scores (0, 0.25, 0.5, 1.0) based on the correctness of the generated text and its alignment with the input prompt. The benchmark includes six evaluation categories: Stepwise or Symbolic Visualization, App & Web UI Simulation, Everyday Digital Moments, Cinematic or Presentation Scenes, Math-Related, and Multilingual (Chinese). Ten text-to-video models (both closed- and open-source) are compared. The reported results show that all models achieve ≤0.5 across all categories, highlighting clear limitations in current text rendering stability.

**Strengths:**

1. The paper addresses an increasingly important problem — evaluating the quality of generated text in videos. As video generation models approach commercial-level quality, consistent and accurate text rendering becomes critical, especially considering the high inference cost of such models.
2. The proposed benchmark covers multiple categories and organizes prompts accordingly, providing a structured view of text generation capabilities across different contexts.
3. The relatively small benchmark size (73 prompts) makes it efficient and cost-effective for comparing different models.

**Weaknesses:**

1. The main limitation is the absence of automatic evaluation. While some aspects of video quality indeed require human judgment, modern OCR methods — even open-source ones — can perform robust text extraction and recognition. Limiting the evaluation solely to human annotation seems unjustified. If human ratings significantly outperform automated metrics, this difference should have been demonstrated empirically.
2. The benchmark comparisons rely on relatively outdated models. Given the rapid progress in text-to-video generation, including publicly available models such as Veo 3 or Kling 2.x would have made the study more relevant and up-to-date.

**Questions:**

1. Why was there no comparison between human and automated (OCR-based) evaluation methods?
2. How do the authors justify that 73 prompts are sufficient to obtain stable and representative evaluation results?

---

### Official Review · Reviewer_3kLT · 2025-10-30

**Soundness:** 2
**Presentation:** 2
**Contribution:** 2
**Rating:** 2
**Confidence:** 3

**Summary:**

This paper presents T2VTextBench, a human evaluation benchmark for assessing on-screen text generation capabilities in text-to-video models. The authors evaluate 10 T2V systems across 73 prompts covering scenarios like UI simulation, cinematic scenes, and mathematical content. Results show that current models struggle significantly with accurate text rendering and temporal consistency. Models demonstrate particular difficulties with geometric text transformations and random character sequences, while performing moderately better on single words. The evaluation reveals a positive correlation between model cost and generation quality, and highlights substantial performance variance across different prompt categories, suggesting that text manipulation remains a critical challenge for modern text-to-video generation systems.

**Strengths:**

This paper evaluates the text-rendering capabilities of existing video generation models and provides insights that may be useful.

**Weaknesses:**

1. The number of test prompts is unclear and appears very limited (only 73 prompts), which significantly weakens the contribution and indicates insufficient workload.
2.  The prompts contain rich textual content, but the paper lacks detailed statistical analysis, such as the average number of words or characters per prompt and text length distribution.
3.  The paper relies entirely on human evaluation conducted by only three volunteers, leading to low reliability with potential subjective errors and evaluation bias.
4.  The prompts are divided into six categories, but the number of prompts in each category is not specified. An unbalanced distribution across categories may affect evaluation fairness and the reliability of conclusions.
5.  The explanation in Observation 4.5 is unconvincing: the paper claims "text-to-video models primarily memorize textual training data at the word level," but normal sentences also require word-level generation. In other words, random word prompts essentially form sentences as well. Following this logic, short sentences with few words should also be generated well since they are word-level as well. So why do random word prompts outperform normal sentence prompts? This explanation lacks persuasiveness.
6.  The paper lacks qualitative comparison results showing different models' outputs for the same prompt.

**Questions:**

See the weaknesses.

---

### Official Review · Reviewer_3FeR · 2025-10-30

**Soundness:** 2
**Presentation:** 2
**Contribution:** 2
**Rating:** 2
**Confidence:** 3

**Summary:**

This paper introduces a novel benchmark designed to assess the capacity of video-generation models for textual control—specifically, their ability to render accurate on-screen text such as captions and mathematical formulas. The evaluation, grounded in human judgment, spans both open-source and commercial models.

**Strengths:**

1、The paper presents a timely and focused investigation into how accurately video diffusion models can render on-screen text.

2、The benchmark covers a comprehensive set of models, spanning both open-source and commercial offerings.

3、The test scenarios are well-differentiated, encompassing digital displays, cinematic titles, mathematical formulas, and multilingual text.

**Weaknesses:**

1、The paper’s core weakness is poor verifiability and scalability: the evaluation is produced only by manual inspection, so inter-annotator disagreement is inevitable, and future models cannot be reproducibly benchmarked against these scores.

2、Only three graduate/undergraduate students served as judges; this tiny pool introduces high sampling variance and undermines statistical confidence.

3、The assessment protocol is overly coarse—just four ordinal labels—with no separate dimensional scoring for legibility, font fidelity, layout accuracy, or temporal consistency.

4、The evaluation scope is overly narrow: the paper offers no evidence that “on-screen” text rendering presents a harder or qualitatively different challenge than general text rendering, so the need for a separate benchmark remains unsubstantiated.

**Questions:**

Please see the weakness above.

---

### Official Review · Reviewer_7RfS · 2025-11-03

**Soundness:** 2
**Presentation:** 2
**Contribution:** 2
**Rating:** 4
**Confidence:** 3

**Summary:**

This paper presents T2VTextBench, the first human-evaluation benchmark for assessing on-screen text fidelity and temporal consistency in text-to-video generation models. The benchmark targets an important and underexplored challenge. how well text-to-video systems render precise textual elements such as captions, signs, or formulas, which are critical for real-world applications in advertising, education, and entertainment. The authors evaluate ten state-of-the-art models, covering both open-source and commercial systems, and identify substantial gaps in current methods’ ability to maintain legible and consistent on-screen text across frames.

**Strengths:**

1.Timely and valuable problem:
The paper addresses a highly practical and important problem that has received limited attention in the current wave of text-to-video research. Ensuring text accuracy in generated videos is crucial for many downstream applications, especially in advertising and other professional creative contexts.

2.Human-centered evaluation:
The idea of human evaluation for assessing text legibility and temporal consistency is meaningful and complements existing automatic metrics.

3.Benchmark initiative:
Establishing a benchmark for this task helps highlight a specific limitation of current systems and provides a direction for future research.

**Weaknesses:**

1.Limited evaluation scale:
The current benchmark includes only 73 prompts, which seems too small to be representative or authoritative. It is unclear how these 73 prompts were designed. What dimensions (e.g., text complexity, motion type, background variation) were considered, and how they were categorized. A more systematic design or justification of prompt selection is needed to improve the credibility of the evaluation.

2.Evaluator expertise:
Only three annotators participated in the human evaluation, which is a very limited number. Moreover, it is not clear whether these evaluators have relevant expertise. Considering the potential applications in advertising or design, involving professional reviewers such as graphic designers, typographers, or creative professionals would make the evaluation more reliable and meaningful.

3.Example quality and visualization:
The examples shown in Figure 4 appear too simple and do not reflect the complexity of realistic advertising or instructional video scenarios. Figures in Figure 6 and the Appendix are too small to read comfortably, especially when the focus is on text fidelity and clarity. Enlarging these figures or including zoomed-in examples would make the results much easier to interpret.

With such a small dataset and limited number of annotators, it is difficult to generalize the conclusions to real-world scenarios. A larger-scale study or an expanded benchmark would strengthen the impact of this work.

**Questions:**

see weakness

---

### Note · Authors · 2025-11-27

**Comment:**

We would like to sincerely thank all the reviewers for providing constructive feedback on our paper. We will carefully address these comments in our next version. After careful consideration, we have decided to withdraw this paper.

**Withdrawal Confirmation:**

I have read and agree with the venue's withdrawal policy on behalf of myself and my co-authors.